# Proteomic Analysis of Zn Depletion/Repletion in the Hormone-Secreting Thyroid Follicular Cell Line FRTL-5

**DOI:** 10.3390/nu10121981

**Published:** 2018-12-14

**Authors:** Barbara Guantario, Angela Capolupo, Maria Chiara Monti, Guido Leoni, Giulia Ranaldi, Alessandra Tosco, Liberato Marzullo, Chiara Murgia, Giuditta Perozzi

**Affiliations:** 1Research Centre for Food and Nutrition, CREA, Via Ardeatina 546, 00178 Rome, Italy; barbara.guantario@crea.gov.it (B.G.); giulia.ranaldi@crea.gov.it (G.R.); 2Department of Pharmacy, Division of Chemistry & Chemical Technologies “Luigi Gomez Paloma”, University of Salerno, Via Giovanni Paolo II, 132 84084 Fisciano (SA), Italy; angela.capolupo@merckgroup.com (A.C.); mcmonti@unisa.it (M.C.M.); 3Nouscom, via di Castel Romano 100, 00128 Rome, Italy; g.leoni@nouscom.com; 4Department of Pharmacy Biomedical Division “Arturo Leone”, University of Salerno, Via Giovanni Paolo II, 132 84084 Fisciano (SA), Italy; tosco@unisa.it; 5Department of Medicine, Surgery and Dentistry “Schola Medica Salernitana”, University of Salerno, Via Salvador Allende, 84081 Baronissi (SA), Italy; marzullo@unisa.it; 6Department of Nutrition, Dietetics and Food, Monash University, Melbourne, VIC 3168, Australia

**Keywords:** endocrine tissues, metal ion, zinc transport, ribosomes, calcium channels

## Abstract

Zinc deficiency predisposes to a wide spectrum of chronic diseases. The human Zn proteome was predicted to represent about 10% of the total human proteome, reflecting the broad array of metabolic functions in which this micronutrient is known to participate. In the thyroid, Zn was reported to regulate cellular homeostasis, with a yet elusive mechanism. The Fischer Rat Thyroid Cell Line FRTL-5 cell model, derived from a Fischer rat thyroid and displaying a follicular cell phenotype, was used to investigate a possible causal relationship between intracellular Zn levels and thyroid function. A proteomic approach was applied to compare proteins expressed in Zn deficiency, obtained by treating cells with the Zn-specific chelator *N*,*N*,*N*′,*N*′-tetrakis (2-pyridylmethyl) ethylene-diamine (TPEN), with Zn repleted cells. Quantitative proteomic analysis of whole cell protein extracts was performed using stable isotope dimethyl labelling coupled to nano-ultra performance liquid chromatography-mass spectrometry (UPLC-MS). TPEN treatment led to almost undetectable intracellular Zn, while decreasing thyroglobulin secretion. Subsequent addition of ZnSO_4_ fully reversed these phenotypes. Comparative proteomic analysis of Zn depleted/repleted cells identified 108 proteins modulated by either treatment. Biological process enrichment analysis identified functions involved in calcium release and the regulation of translation as the most strongly regulated processes in Zn depleted cells.

## 1. Introduction

The micronutrient zinc (Zn) is actively transported across biological membranes by lipid bound specialized transport proteins belonging to the SLC30 (ZIP) and SLC39 (ZnT) families [1]. As integral components of hundreds of enzymes, transcription factors, and structural proteins, Zn ions have a wide range of functions inside cells, but free, non-protein bound ions can also contribute a regulatory role through signal transduction [2]. It is not surprising therefore that the human Zn proteome was predicted to represent about 10% of the total human proteome on the basis of the occurrence of Zn binding motifs [3]. Although a specific role for Zn ions has been demonstrated in several biological processes, the emerging picture is not yet exhaustive, especially in relation to pathologies, and further insights could be very helpful in the clinical setting. To identify pathways potentially affected by Zn deficiency/excess and playing important roles in disease development, we have recently performed an in silico approach which extended the known Zn proteome by also including the Zn-binding protein interactome. The resulting Zn protein interaction network (ZNP) included proteins involved in several biological processes that are likely to be affected by Zn availability in health and disease, among which are protein localization and transport, as well as hormone secretion and hormone-dependent physiological processes [4].

Focusing on this latter aspect, Zn was shown to be required for proper functioning of the secretory pathway, as both a catalytic and structural component of the secretome. The activity of specific transporters of the ZnT family, regulating intracellular Zn compartmentalization, was in fact reported to regulate hormone secretion [5]. Involvement of ZnTs was shown in the activation of ectoenzymes either as heterodimers, as is the case for ZnT5/ZnT6, or as homodimers, as shown for ZnT4 and ZnT7 [5]. Tissue specific expression of Zn transporters represents a second level of regulation of Zn-dependent processes. We have previously shown that the ZnT8 transporter, primarily expressed by pancreatic beta cells where it regulates insulin secretion, is also present in specific endocrine cell types of pituitary, adrenal, and thyroid glands, suggesting a more general role in hormone secretion [6]. Experimental evidence has been provided for a role of Zn in thyroid homeostasis: serum concentrations of both triiodothyronine and thyroxine were decreased in rats fed a Zn deficient diet [7]; and, Zn supplementation was reported to improve serum thyroid hormone levels in goitrous patients [8]. Observational evidence collated in a systematic review suggested a positive association between Zn status and thyroid hormones [9]. However, the biological mechanisms and the possible clinical consequences of the disruption of Zn dependent processes in thyroid cells are still unknown [10,11]. Omic-approaches and systems biology were proven effective to detail specific genes and regulatory molecules in pancreatic endocrine β-cell function and dysfunction [12]. Proteomics was also applied to gather mechanistic insights into thyroid pathophysiology, resulting in the identification of biomarker proteins differentially expressed in benign/malignant thyroid lesions [13,14]. We therefore chose to employ a proteomic approach to investigate the molecular basis of the role of Zn and its specific transporters in thyroid hormone metabolism. Our aim was to clarify a possible causal relationship between intracellular Zn levels and thyroid function, by comparing cells whose intracellular free Zn pool had been depleted by treatment with the specific Zn chelator TPEN, and cells in which Zn levels were restored by the addition of ZnSO_4_ to the culture medium. To address these unsolved issues and acquire a deeper understanding of endocrine processes dependent on intracellular Zn homeostasis, we chose the in vitro thyrocyte model FRTL-5, derived from Fischer rat primary thyroid cells and displaying a normal follicular cell phenotype in terms of growth properties, thyroid-stimulating hormone (TSH) dependence, expression of TSH receptor, and ability to synthesize and secrete thyroglobulin. 

In this paper, we report the results of characterization of the Zn-related molecular and physiological phenotype of FRTL-5 cells, and a comparison between their baseline proteomic signature with the corresponding profiles in marginally Zn deficient thyroid follicular cells, as well as upon restoring intracellular Zn stores.

## 2. Materials and Methods 

### 2.1. Cell Culture Maintenance and Drug Treatment

The Fischer rat thyroid follicular FRTL-5 cell line was obtained from the American Type Culture Collection (CRL-1468). Cells were cultured in Ham’s F12 medium supplemented with 5% fetal bovine serum (FBS), 100 U/L penicillin, 100 µg/L streptomycin, and a six-hormone mixture composed of: 10 µg/mL insulin, 10 nM hydrocortisone, 5 µg/mL apo-transferrin, 10 ng/mL somatostatin, 0.01 I.U./mL thyrotropic hormone (TSH), and 10 ng/mL Gly-His-Lys acetate salt. Medium was replaced every two days. After reaching 70–80% confluence, cells were either sub-cultivated or used for the experiments [15].

To induce marginal Zn deficiency, FRTL-5 cells were incubated in experimental medium containing *N*,*N*,*N*′,*N*′-tetrakis (2-pyridylmethyl) ethylene-diamine (TPEN) at the concentration and for the time indicated, in the presence or absence of FBS and the six-hormone mixture. Following TPEN removal, cells were exposed to fresh medium or to medium containing different concentrations of ZnSO_4_. All reagents were from Sigma Aldrich, Milan, Italy, unless otherwise stated.

### 2.2. Fluorescence Imaging and Assays

Immunofluorescence experiments were performed by standard protocols on cells fixed either in methanol at −20 °C, or in 2% paraformaldehyde with permeabilization in 0.1% TRITON X-100, followed by staining with primary antibodies and corresponding secondary antibodies. The following mouse monoclonal antibodies were used: α-adaptin (AP2), β1 and β2 adaptins, β-COP (Sigma Aldrich, Milan, Italy), gm130 (Santa Cruz Biotechnology, Dallas, TX, USA), TGN46 (Abcam, Cambridge, UK), transferrin receptor (Zymed, Thermo Fischer Scientific, Rodano (Mi), Italy), and rabbit polyclonal ZnT8 [6]. Secondary conjugated antibodies were goat anti-mouse IgG FITC and goat anti-rabbit IgG TRITC (tetramethylrhodamine) (Jackson Immunoresearch Laboratories, Cambridgeshire, UK). Intracellular zinc was imaged with 1 µM FluoZin-3-AM (Molecular Probes, Invitrogen Waltham, MA, USA), a cell-permeant dye with high affinity to zinc ions (KD = 15 nM), in samples fixed in paraformaldehyde without permeabilization. Preparations were mounted using ProLong Gold antifade Reagent (Molecular Probes, Invitrogen, Waltham, MA, USA). For nuclear staining, 300 nM 4′,6-diamidino-2-phenylindole dihydrochloride (DAPI) (Sigma-Aldrich, Milan, Italy) was added directly to the mounting medium. Specimens were analyzed using an inverted laser-scanning confocal microscope with a 40 × oil immersion objective (LSM 700; Carl Zeiss, Jena, Germany).

For fluorescence assays, FRTL-5 cells were grown in 96-well black polystyrene microplates to confluence, and incubated with experimental medium containing TPEN or ZnSO_4_ for 2 h at the indicated concentration. At the end of the incubation time, cells were washed with 1X PBS, and 1 µM FluoZin-3-AM was added for 30 min. Intracellular fluorescence was measured in a microplate UV reader at Ex/Em: 494/516 nm (Tecan, Infinite M200, Mannedorf, Switzerland).

### 2.3. Immunoblotting

Cells were harvested in cold lysis buffer (1% SDS in 1 × PBS and protease inhibitor cocktails), and samples were stored O/N at −20 °C. Samples were then heated at 80 °C for 5 min and centrifuged at 15,000× *g* for 10 min. The supernatants were transferred into new tubes for protein quantification and electrophoresis. For thyroglobulin secretion, culture media were collected and centrifuged at 15,000× *g* for 15 min; supernatants were transferred into new tubes and stored at −80 °C. Aliquots of cell lysates (20 µg of total protein) and culture media (20 µL) were dissolved in Laemmli buffer, heated at 80 °C for 5 min, fractionated by 4–20% SDS polyacrylamide gel electrophoresis (SDS–PAGE), and then transferred to nitrocellulose filter. Membranes were incubated with the following antibodies: goat polyclonal anti-XIAP (R&D System, R&D System, Minneapolis, MN, USA), rabbit polyclonal anti-ZnT2 (H-40), goat polyclonal anti-ZnT4 (N-17) (Santa Cruz Biotechnology, Dallas, TX, USA), rabbit polyclonal anti-ZnT8 [6], mouse monoclonal anti-thyroglobulin (Santa Cruz Biotechnology, Dallas, TX, USA), and mouse monoclonal anti-α-tubulin (MP Biomedical, Santa Ana, CA, USA). Proteins of interest were detected with horseradish peroxidase-conjugated secondary antibodies (Cell Signaling Technology, NL, Danvers, MA, USA) and enhanced chemiluminescence reagent (Euroclone, Pero, Mi, Italy). Images were acquired with the CCD camera detection system Las4000 Image Quant (GE Health Care, Milan, Italy).

### 2.4. Proteomic Analysis by Stable Isotope Dimethyl Labeling (DML)

#### 2.4.1. Labeling Reaction

Eighteen samples of protein lysates obtained from cells incubated with TPEN (six samples named TPEN) or without (six samples named CTR, control) or exposed to medium containing ZnSO_4_ after TPEN removal (six samples named REC, recovery ) were pooled in three different batches and protein content was quantified by Bradford analysis; 25 µg of each sample was dried, solubilized in Laemmli buffer, and loaded on a 12% acrylammide SDS-PAGE, run for 5 min to remove low mass impurities from the samples. One large band for each sample was then excised from the gel and digested as reported by Shevchenko et al. [16]. The obtained peptides were labeled following an in-solution protocol, as reported by Boersema et al. [17]. Briefly, each sample was reconstituted in 400 µL of 100 mM triethyl ammonium bicarbonate (TEAB) and 10 µg of Glu-Fibrinogen peptide was carefully added to each sample as the internal standard. A total of 160 µL of 4% formaldehyde was added to the TPEN sample, and 4% D-formaldehyde was separately added to the CTR and REC samples. Then, 16 µL of 0.6 M NaBH_3_CN was separately added to the TPEN and REC samples and 16 µL of 0.6 M NaBD_3_CN was added to the CTR sample. The solutions were stirred for 1 h at 20 °C and quenched by the addition of 1% ammonia solution. After acidification, samples were accurately mixed in a 1:1 ratio, dried, and reconstituted. One-third of the sample was subjected to zip tip purification, as reported by the manufacturer (Merck Millipore, Darmstadt, Germany). 

#### 2.4.2. Mass-Spectrometry Analysis

Each peptide sample was dissolved in formic acid (FA, 10%) and 5 µL was injected into a nano-ACQUITY UPLC system (Waters, Milford, MA, USA). Peptides were separated on a 1.7 mm BEH C_18_ column (Waters, Milford, MA, USA) at a flow rate of 300 nL/min. Peptide elution was achieved with a linear gradient (solution A: H_2_O (95%), CH_3_CN (5%), FA (0.1%); solution B: CH_3_CN (95%), H_2_O (5%), FA (0.1%)); 15–50% B over 180 min). MS and MS/MS data were acquired with an LTQ-Orbitrap XL (ThermoFisher, Waltham, MA, USA). The fifteen most intense doubly and triply charged peptide ions were chosen by the Xcalibur software version 4.0 (ThermoFisher, Waltham, MA, USA.) and fragmented. The resulting MS data were processed to generate peak lists for protein identifications.

#### 2.4.3. Bioinformatics Analysis

Database searches were carried out using MaxQuant (version 1.5.5.1., Max-Planck-Gesellschaft, Munchen, Germany http://www.maxquant.or), with the Andromeda search engine against the Swiss Prot database (558898 entries), with a precursor mass tolerance of 20 ppm and fragment mass deviation of 0.8 Da. The search included variable modifications of methionine oxidation and fixed modification of cysteine carbamidomethylation. Trypsin was set as the specific proteolytic enzyme. Peptides with a minimum of five amino acids and a maximum of two missed cleavages were allowed for the analysis. For both peptide and protein identification, the cut-off false discovery rates (FDR) were set at 0.01. Doublets were selected as the quantification mode with the dimethyl Lys 0 and N-term 0 as the light label, namely TPEN; dimethyl Lys 4 and N-term 4 as the intermediate label, namely REC; and dimethyl Lys 8 and N-term 8 as the heavy label, namely CTR. Re-quantify mode was enabled. Default settings were used for all other parameters in MaxQuant.

Modulated proteins in each comparison with a log2 fold change (FC) ± 1 were identified. Clustering analysis was performed with the ward method by using the Euclidean distance measure to identify relevant clusters. Biological pathways significantly enriched by modulated proteins were identified using the DAVID server [18]. Biological processes annotated in the DAVID direct GO database enriched with a *p*-value less than 0.05 were considered as significantly enriched.

### 2.5. Statistical Analysis

Prior to the analysis, a normal distribution and homogeneity of variance of all variables were controlled with Shapiro-Wilk and Levene’s tests, respectively. For immunoblotting experiments, statistical significance was evaluated by Welch one-way ANOVA, followed by a Tamhane test. Statistical univariate analysis was performed with the Microsoft Office Excel 2011 upgraded with XLSTAT (version 4 March 2014, Microsoft, Redmond, WA, USA). Differences with *p*-values < 0.01 were considered significant. Fluorescence experiments were analysed by a paired Student *t*-Test, and differences with *p*-values < 0.001 were considered significant.

## 3. Results

### 3.1. Characterization of Zn Metabolism in the FRTL-5 Cell Line

The Fischer rat thyroid follicular FRTL-5 cell line is a functional, immortalized thyroid cell line displaying some of the functions of differentiated thyrocites, such as TSH-dependent proliferation and differentiation, iodine uptake, thyroglobulin, and thyroperoxidase gene transcription [19]. To verify whether FRTL-5 cells could represent a good in vitro model for studying Zn metabolism in thyrocites, we initially defined the culture conditions, allowing the expression of functional markers of follicular, as well as Zn-related, phenotypes. Cells were therefore grown in the presence and absence of serum and a specific cocktail of hormones (see methods for details). The supernatant was collected 2, 4, 8, and 24 h after medium change and analyzed by Western blotting for expression of the thyroid hormone precursor thyroglobulin. The results in Figure 1A show detectable thyroglobulin secretion after 4 h of culture in the presence of serum and hormones, while in the absence of serum, detection of thyroglobulin required 24 h cell growth, despite the presence of hormones. Panel B shows Western blotting of total cell lysates with specific antibodies against the three vesicular transporters ZnT2, T4, and T8, which were previously associated with the secretory pathway in other cell types [20,21,22].

Since ZnT8 associates with secretory vesicles in other cell types [6,23,24], we further investigated its intracellular localization in thyrocites by double staining cells with anti-ZnT8 antibodies and with the zinc specific fluorescent probe Fluozin-3. 

As shown in Figure 2, ZnT8 localized to an intracellular vesicular compartment in FRTL-5 cells (panel A). Co-staining with Fluozin-3 highlighted vesicle-like structures localized in the perinuclear region of the cells (panel B). These vesicles, however, did not appear to overlap with ZnT8-containing vesicles (panel C), whose identity we sought to identify by immunofluorescence staining with specific antibodies against the following vesicular markers of the secretory pathway: (I) anti-β1β2 adaptins and α-adaptin, the larger subunits of the AP-2 adaptor complex, which stain clathrin-coated vesicles; (II) anti-β-COP, which labels the periphery of the Golgi complex; (III) the anti-gm130 and anti-TGN46 markers of the trans-Golgi network; and (IV) anti-transferrin receptor to label recycling endosomes. Figure 3 shows that ZnT8 localized in abundant vesicles scattered throughout the cytoplasm in FRTL-5 cells, but no clear co-localization could be detected with any of the labeled compartments.

To determine the Zn transport capacity in this cell model and quantify intracellular free Zn levels in response to different treatments, we set up a fluorimetric technique based on Fluozin-3 binding to free Zn ions. FRTL-5 cells were either Zn-depleted by adding increasing concentrations to the culture medium of the specific Zn chelator TPEN, which we and others routinely use to induce conditions of intracellular mild Zn deficiency in other cell lines [25,26,27,28], or challenged with increasing concentrations of ZnSO_4_. The results in Figure 4 show that Zn uptake in these cells is very efficient and dose-dependent in the 10–160 µM range of ZnSO_4_, while TPEN treatment resulted in increasing levels of the chelation of intracellular Zn ions, as shown by the significantly decreased fluorescence that reached about 30% of control levels at the highest concentration used (100 µM).

### 3.2. Conditions for Proteomic Analysis

Having established that FRTL-5 cells are capable of efficiently importing Zn ions from the experimental medium and properly storing them in an intracellular vesicular compartment, we then sought to determine a minimum set of conditions which could enable us to identify specific functions involved in Zn fluxes in thyrocites by a comparison of proteomic profiles in Zn depleted/repleted vs. control cells. To induce Zn depletion, we chose the lowest TPEN concentration which was able to decrease intracellular Zn levels in the Fluozin-3-based fluorescence assay (25 µM for 2 h, Figure 4), further confirming the possible involvement of Zn ions in secretion of the thyroid hormone precursor in follicular cells. Furthermore, cell survival was almost unaffected by 25 µM TPEN treatment for 2 h).

To identify a ZnSO_4_ concentration able to fully revert the effects of the TPEN-induced intracellular Zn deficiency, we monitored changes in the expression levels of XIAP (X-linked inhibitor of apoptosis protein), a regulator of cell survival whose transcription we and others have previously shown to be reduced in response to decreased concentrations of intracellular free Zn in other cell systems [29,30], following 25 µM TPEN treatment for 2 h and the subsequent addition of 25 µM ZnSO_4_ for 24 h (Recovery). The results of Western blot analysis of the corresponding cell lysates with anti-XIAP antibodies, shown in Figure 5A, displayed significantly decreased XIAP protein levels in TPEN treated, marginally Zn deficient FRTL-5 cells. Subsequent Zn repletion restored XIAP expression to control levels after 24 h. Moreover, Zn fluorescence imaging confirmed that TPEN induced Zn deficiency, as well as subsequent rescue following incubation in the presence of added Zn (Figure 5B). We therefore used these as the optimal conditions to investigate the physiological response to Zn depletion/repletion in thyrocites by proteomic analysis.

On the basis of these results, we defined the following two experimental conditions for the proteomic analysis of FRTL-5 cells, which are graphically displayed in Figure 5C: (1) Zn deprivation, corresponding to 2 h incubation in the presence of 25 µM TPEN; and (2) Zn repletion, corresponding to 24 h incubation in the presence of 25 µM ZnSO_4_ after TPEN treatment.

### 3.3. Proteomic Analysis

FRTL-5 cells were treated for proteomic analysis as depicted in Figure 5C. Total proteins were extracted from cells treated as follows: (1) Control cells (CTRL), grown in culture medium containing serum and hormones; (2) Zn deprived cells (TPEN), incubated for 2 h in the presence of 25 µM TPEN; and (3) Zn repleted cells (REC), corresponding to 24 h "recovery" incubation in the presence of 25 µM ZnSO_4_. The results of proteomic analysis identified a total number of 108 proteins as modulated by Zn depletion and/or repletion (Figure 6). Their distribution among the three possible pairwise comparisons is presented in Panel A, while Panel B shows the full list of proteins and the corresponding level of up- or down-regulation in the three different culture conditions. The analysis in Panel A shows that 53 proteins were up- or down-regulated in Zn depleted cells (TPEN) when compared to CTRL cells, while 64 proteins were modulated by Zn repletion (REC) when compared to TPEN treated cells. Interestingly, a cluster of 24 proteins was modulated by both treatments. In this cluster, Zn repletion mainly counteracted the effect of TPEN chelation by regulating the expression in the opposite direction (Appendix A). Most of these proteins (16 out of 24) were in fact upregulated by TPEN treatment (TPEN vs. CTRL) and down-regulated following Zn repletion (REC vs. TPEN) (Figure 6B, Appendix A). The effect of Zn repletion, however, did not appear to be limited to balancing the alterations in protein expression induced by the previous TPEN treatment, as 15 proteins were exclusively modulated by Zn repletion (REC vs. CTRL). Furthermore, 23 proteins whose expression had not been significantly altered by Zn depletion (TPEN vs. CTRL) were specifically and significantly down-regulated by Zn repletion only when compared to their levels in Zn depleted cells (Recovery vs. TPEN) (Appendix A).

To verify whether the modulated proteins in Zn depleted/repleted thyrocites were included in the Zn proteome interaction network that we previously described [4], we performed a preliminary comparative analysis. Figure 6C shows that 59 of the 108 modulated proteins are indeed listed in the ZNP; 11 of them were found to represent Zn-binding proteins, while the remaining 48 were among the Zn binding protein interactors. Figure 6D shows a representative MS spectrum of the peptide 22–30 from HIST1H2AJ; the spectrum represents doubly charged ions of the peptide labeled as light (TPEN) at *m*/*z* of 486.78, as intermediate (REC) at *m*/*z* of 488.80, and as heavy (CTRL) at *m*/*z* of 490.81. The MaxQuant software is able to quantify the ions belonging to the same protein marked with different tags by the MS spectra and to identify the proteins in the database using the related MSMS spectra (Appendix A).

The full list of modulated proteins underwent gene ontology biological process (BP) enrichment analysis (Figure 7, Appendix A). Ten biological processes were estimated as significantly enriched by 17 proteins modulated following Zn depletion (Figure 7A, TPEN vs. CTRL). Enriched BPs include processes involved in calcium release, carbohydrate metabolism, and the regulation of translation. The majority of the corresponding proteins were up-regulated by Zn depletion, and were numerically more represented in molecular functions related to RNA and protein binding (panel B, Appendix A). Only five biological processes were indicated as enriched after the subsequent Zn repletion treatment (Figure 7C, REC vs. TPEN), mainly related to the regulation of cell polarity, cell adhesion, and translation (Appendix A). Modulation of the corresponding proteins was mainly in the direction of down-regulation, and the specifically enriched molecular functions mostly involved GTP binding and hydrolysis, along with ribosomal constituents (Figure 7D, Appendix A).

## 4. Discussion

Zn ions are known to accumulate in intracellular vesicles, which are especially abundant in endocrine cells, where they were shown to play a role in secretion in some cell types [5]. Zn involvement in insulin secretion has been extensively studied in pancreatic β-cells, but a broader role for this metal in endocrine tissues was proposed on the basis of the expression profile of vesicular Zn transporters associated with the secretory pathway [6,22,24]. The ZnT8 transporter, in particular, is necessary for insulin secretion in pancreatic β-cells, and we and others subsequently demonstrated its abundant expression in follicular cells of the thyroid gland, which actively secrete thyroid hormones, suggesting a potential role for Zn in this process [6,10,24,25]. In the present study, we have shown that the three main intracellular Zn transporters associated with the secretory pathway (ZnT2, T4, and T8) [20,22,31] are expressed in the thyroid follicular cell culture model FRTL-5, which actively secretes the thyroid hormone precursor thyroglobulin. In pancreatic β-cells, the ZnT8 transporter was shown to co-localize with zinc and insulin in secretory vesicles [20]. ZnT8 appears to localize to an intracellular vesicular compartment in FRTL-5 cells as well, which made us pursue the hypothesis that it might be involved in a similar mechanism in thyrocites. However, ZnT8 immunolocalization in FRTL-5 cells does not appear to overlap with any intracellular vesicular compartment that we could label with specific markers of the secretory pathway, which led us to exclude the option that this transporter might play a similar role in thyroid hormone secretion. We have also shown that FRTL-5 cells are able to take up zinc ions from the external environment with a very efficient and dose-dependent mechanism that is not saturated up to at least 160 µM ZnSO_4_. Once inside the cell, Zn ions are efficiently accumulated in an intracellular vesicular compartment. Subsequent results of proteomic analysis in Zn depleted/repleted cells, however, did not show the differential expression of any of the ZnTs that were previously associated with the secretory pathway, and therefore, we did not pursue further characterization of their intracellular localization in thyrocites.

A first attempt to identify Zn dependent processes in thyroid cells was based on a proteomic approach. Proteomics was previously applied to the thyroid to identify proteins able to define differences between benign/malignant lesions, with the aim of developing novel biomarkers for the advanced diagnosis of thyroid pathologies [32]. To gain further insight into the mechanisms underlying marginal Zn deficiency in hormone secreting thyroid follicular cells, we identified suitable conditions of Zn depletion/repletion, which could mimic the in vivo condition. Proteomic analysis was then applied to whole cell extracts from control cells, displaying cytoplasmic vesicular Zn fluorescence. Zn depleted cells showed almost undetectable cytoplasmic Zn fluorescence, and reduced levels of XIAP expression and thyroglobulin secretion; in Zn repleted cells, all Zn depletion phenotypes were restored to control levels. The results identified 53 proteins modulated in Zn depleted cells, and 64 proteins which were modulated by the subsequent Zn repletion treatment. Regulation of most proteins in this latter group was indeed in the opposite direction, thus indicating that Zn addition to the culture media was effective in restoring alterations induced by Zn depletion. However, 15 proteins appeared to be exclusively modulated by Zn repletion when compared to control cells, highlighting phenotypic differences between the initial state profiled in CTRL cells and the “restored” state that we detect following Zn repletion in TPEN treated cells. This result suggests that Zn depleted cells do not fully revert to the metabolic state of control cells upon Zn repletion. This is even more apparent when looking at differential expression data for these proteins (Appendix A), as we can see that most of them were indeed modulated by Zn depletion, but the fold changes were below the threshold. 

The major biological processes affected by alterations of Zn homeostasis in thyrocites were related to calcium release, carbohydrate metabolism, and the regulation of translation in Zn depletion, while the regulation of cell polarity, cell adhesion, and translation (ribosomal constituents) were modulated by Zn repletion. In general, Zn depletion resulted in up-regulation of protein expression, while Zn repletion exerted a down-regulating effect. The majority of the modulated processes, if not all, in both Zn deficient and Zn repleted cells, can be directly or indirectly related to the ribosomal pathway and to calcium signalling. The significant effect on ribosome components confirmed the key role that Zn plays in the biogenesis of this organelle. The eukaryotic ribosome is made of four ribosomal RNAs and 79 proteins, and its biogenesis is a demanding process in terms of energy and was shown to be affected by perturbations in nutrients’ supply [33]. Zn plays a direct role in ribosome biogenesis, being essential for the stability and activity of RNA polymerase I, the multi-subunit enzyme specialised in the synthesis of rRNAs, which are the core components of ribosomes. Moreover, several ribosomal proteins bind Zn. It was proposed that intracellular Zn availability could be maintained by regulating ribosome biogenesis, as a significant fraction of cellular Zn is used for this process [31]. Transcriptional activities during zinc starvation were reported to follow a hierarchical order. Degradation of the most active cellular transcriptional enzyme, namely RNA polymerase I, couples cellular growth to zinc availability. On the contrary, RNA polymerase II was unaffected, likely to permit the synthesis of proteins needed by the cell for Zn transport [31]. The Zn containing transcription factor ZNF658, which regulates the transcription of rRNAs and of several ribosomal proteins, was identified as the link between ribosome assembly and Zn homeostasis in human cells. Therefore, ribosome biogenesis, regarded as the most Zn-demanding cellular process, is linked through ZNF658 to RNA polymerase I, the most active transcriptional activity of the cell [34]. Until not long ago, the ribosome was considered to contain the same components in every tissue and cell type within a species, but ribosomal proteins are now known to be encoded by variants that can be expressed in a tissue-specific manner [35]. In line with these findings, we confirmed that Zn homeostasis is tightly connected with ribosome biology in the experimental model of cultured thyroid cells, most likely through a complex regulation system that will require more investigation to be unravelled.

A second set of functions that appears strongly regulated by Zn depletion/repletion relates to calcium metabolism. Aside from its role in bone remodelling, Ca controls a wide variety of processes, mainly in the cardiovascular, nervous, and endocrine systems, including muscle contraction, hormone secretion, organelle communication, and cell growth, among others. These processes are all regulated by Ca signalling, pointing at the pivotal role of Ca homeostasis for the proper functioning of cells and tissues [36]. Zinc was shown to move in and out of cells not only through the specific ZIP/ZnT transporters, but also via some of the isoforms of Voltage Activated Calcium Channels (VACC) [36,37]. Although the majority of intracellular Zn plays a structural/regulatory role as an integral component of intracellular proteins, while Ca acts mainly through signal transduction, both of these ions are essential for the functionality of a broad array of cellular functions, and their intracellular homeostasis is tightly regulated at the level of influx and intracellular compartmentalization, requiring a certain level of coordination. In line with this possible Zn-Ca cross-talk, we find that some Ca-channel proteins and Ca-modulated functions are enriched within the pool of up-regulated proteins in Zn depleted cells, while they return to control levels upon the repletion of intracellular Zn stores. The interplay between these two crucial elements within cells is likely to affect the same key pathways controlling the equilibrium between healthy and dysfunctional states in chronic diseases [36,38].

Notably, the pathways identified in the present work extend to thyrocites the validity of our in silico approach on the Zn proteome interaction network (ZNP, [4]), as more than half of the proteins modulated in Zn depletion/repletion conditions are listed in the ZNP. The fact that 48 of these 59 proteins appear to be interactors of Zn-binding proteins confirms the importance of considering the interactome as an integral part of the network. Due to the intrinsic nature of in silico approaches, our previous work could not specifically attribute the identified Zn modulated pathways to specific cells and tissues, but rather set an overarching frame of Zn modulated pathways irrespective of cell type. The present work goes one step further by pointing at a subset of those pathways which appear especially vulnerable to the alteration of intracellular Zn homeostasis in a specific cell type, i.e., follicular thyroid cells. Identification of Zn regulated proteins in thyrocites can thus drive further research on tissue-specific processes that can be altered by Zn status, with important physiopathological implications in health and disease as the response to Zn deficiency in this endocrine tissue was previously unexplored at the molecular level.

## 5. Conclusions

FRTL-5 cells represent a suitable model to study the role of Zn in thyroid follicular cell physiology. Analysis of the Zn-related cell proteome highlighted ribosome assembly and calcium–dependent processes as the main functions whose molecular components are altered by intracellular Zn status in thyroid derived cells. These insights can be further exploited to explore the predisposing effect of Zn deficiency towards the onset of chronic disease at the molecular level.

## Figures and Tables

**Figure 1 nutrients-10-01981-f001:**
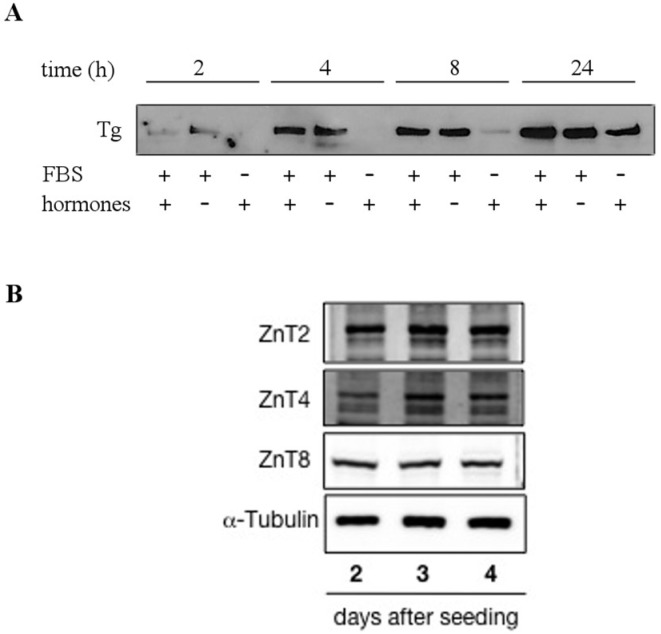
Secretion of thyroglobulin and expression of ZnT transporters in FRTL-5 cells. (**A**) Confluent cells were grown for 24 h in the presence (+) or absence (-) of serum (FBS) and/or a hormone cocktail. Supernatants were collected after 2, 4, 8, and 24 h and analyzed by Western blotting with anti-thyroglobulin (Tg) antibodies. (**B**) Western blot analysis of whole cell lysates, collected at 2, 3, and 4 days after seeding, with anti-ZnT2, ZnT4, and ZnT8 antibodies. Cells were grown in the presence of FBS and hormones.

**Figure 2 nutrients-10-01981-f002:**
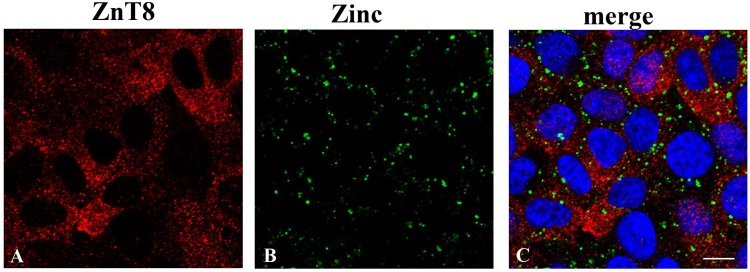
Distinct localization of the zinc transporter ZnT8 and vesicle-like structures containing labile zinc in FRTL-5 cells. Cells grown in standard culture conditions were co-stained with: (**A**) anti ZnT8 polyclonal antibody and (**B**) FluoZin-3-AM. (**C**) Merged image of panels (**A**) and (**B**) with DAPI staining of nuclei. All images represent optical sections obtained by confocal acquisition. Size bar corresponds to 10 µM.

**Figure 3 nutrients-10-01981-f003:**
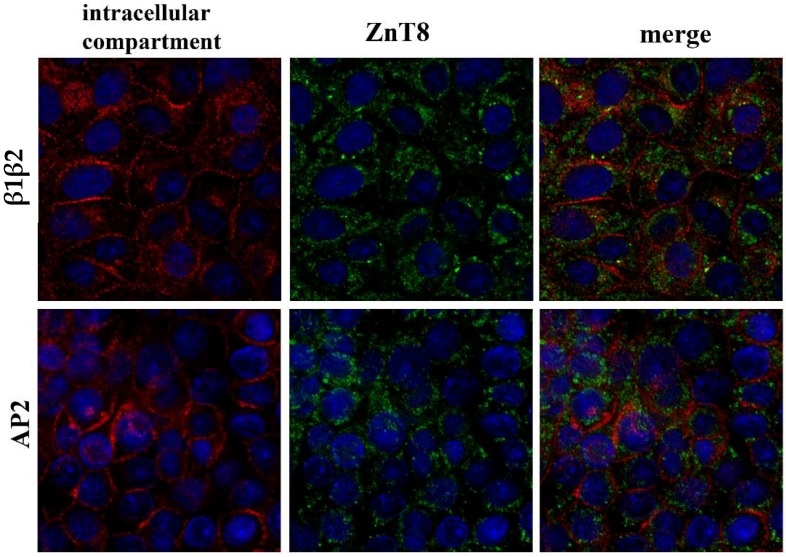
Intracellular localization of the zinc transporter ZnT8 and markers of intracellular vesicular compartments. Each row of panels contains the individual and merged images of double fluorescent staining with anti-ZnT8 (green fluorescence) and a distinct antibody against specific markers of intracellular compartment indicated on the left (red fluorescence). Markers: β1β2, AP2 (plasma membrane—clathrin vesicle trafficking); TGN 46 (Trans-Golgi Network); gm130 (Golgi compartment); β-COP (Golgi compartment—Endoplasmic Reticulum); TF rec (Transferrin receptor—recycling endosomes). Cell nuclei were labeled with DAPI (blue fluorescence). Images represent optical sections obtained by confocal acquisition. Size bar corresponds to 10 µM.

**Figure 4 nutrients-10-01981-f004:**
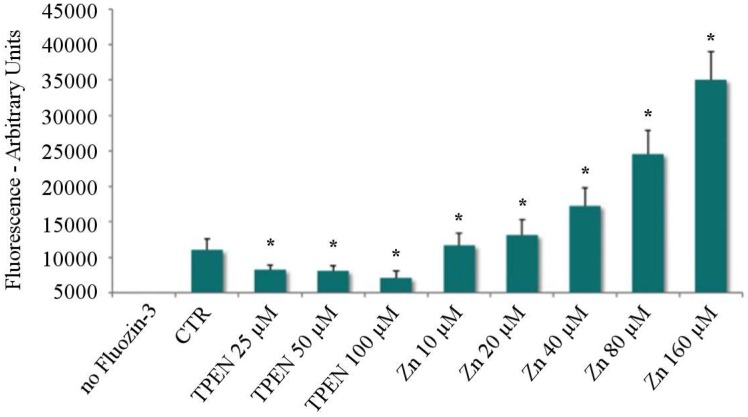
Intracellular Zn depletion by treatment of FRTL-5 cells with increasing concentrations of the Zn-specific chelator TPEN. Fluorimetric analysis of intracellular free Zn levels in FRTL-5 cells under standard culture conditions. Each column represents the average ± SD of three independent experiments. The arbitrary unit fluorescence reflects Zn binding by the Zn-specific fluorescent probe Fluozin-3 (*p* < 0.01) for TPEN and Zinc treatments—Paired Student’s *t*-Test. Asterisks “*” represent significant differences among treatments (*p* < 0.01).

**Figure 5 nutrients-10-01981-f005:**
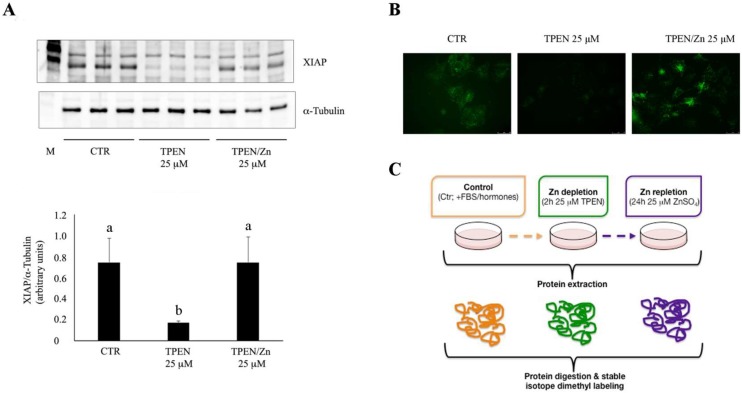
Effect of Zn depletion/repletion on XIAP expression in Zn depleted/repleted FRTL-5 cells. (**A**) Top panel: immunoblotting of whole cell lysates with anti-XIAP antibodies. Bottom panel: quantitative analysis of two independent experiments performed in triplicate. Each column represents the average of XIAP band intensities normalized to the corresponding values for α-tubulin. Data are expressed as mean ± SD. Statistical significance was evaluated by Welch one-way ANOVA followed by a Tamhane test; different letters indicate significant differences (*p* < 0.01). (**B**) Intracellular Zn staining with Fluozin-3 in FRTL-5 cells treated as described in panel C (CTR = control). (**C**) Study design: Confluent FRTL-5 cells were treated for 2 h with 25 µM TPEN to induce marginal Zn deficiency and then incubated with 25 µM ZnSO_4_ for 24 h to replete intracellular Zn stores. After treatments, total proteins were extracted, digested, and analyzed by mass spectrometry.

**Figure 6 nutrients-10-01981-f006:**
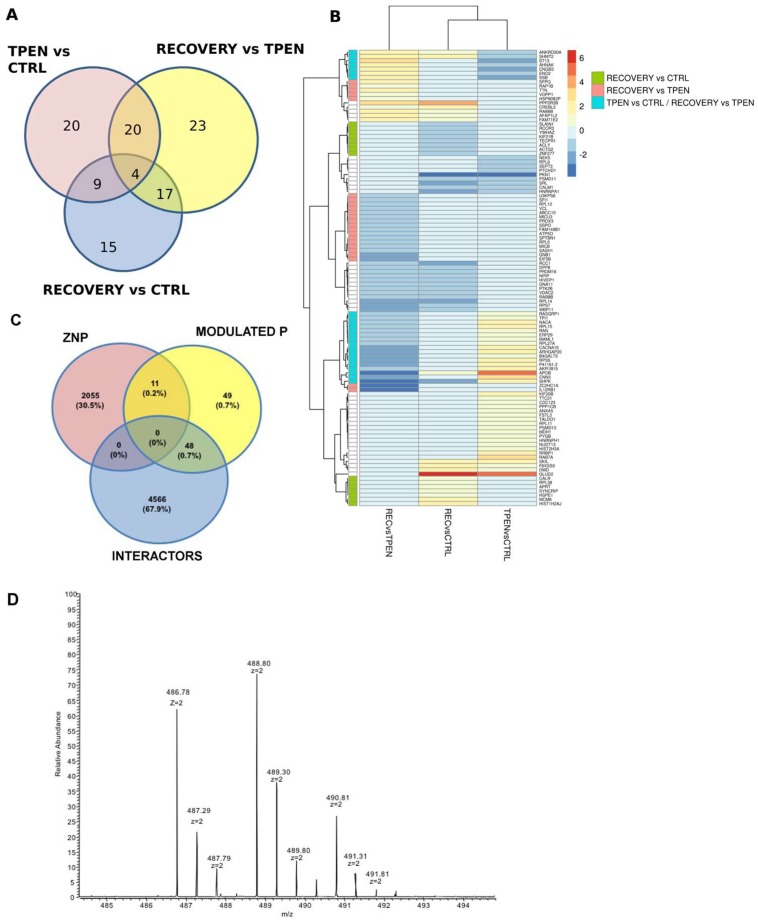
(**A**) Venn diagram comparing the distribution of modulated proteins in the three different conditions. (**B**) log2 fold change for each modulated protein visualized as a heatmap. Green, light red, and light blue boxes on the left side refer to the relevant clusters discussed in the text. (**C**) Venn diagram showing the interaction among three domains of zinc-related proteins: ZNBP, ZiNc Binding Proteins listed in the Zn proteome interaction network [4]; ZNThy, proteins modulated in thyroid by Zn depletion/repletion in this work; INT: Zinc binding protein INTeractors listed in the Zn proteome interaction network [4]. (**D**) Representative doubly charged ions of a peptide belonging to an identified protein labeled as light, intermediate, and heavy.

**Figure 7 nutrients-10-01981-f007:**
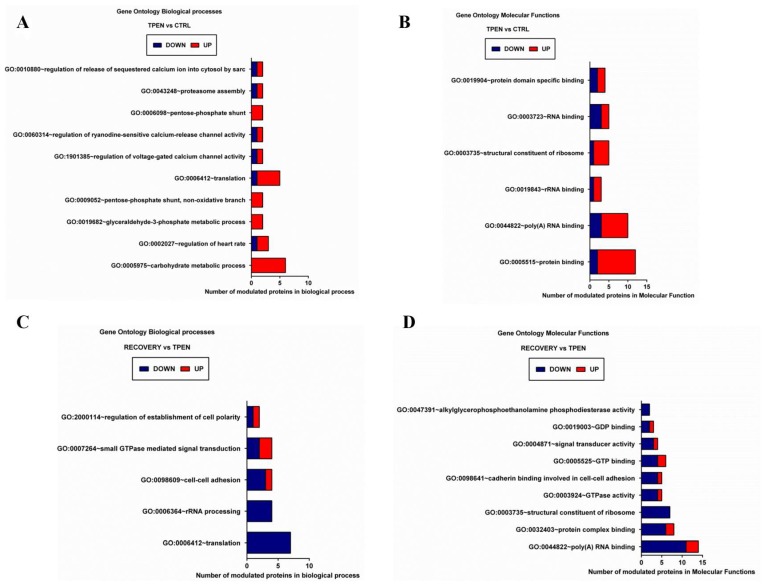
Gene ontology biological processes (**A**,**C**) and molecular functions (**B**,**D**) significantly enriched by proteins modulated in Zn depletion (TPEN (Zn deprived cells) vs. CTRL (Control cells), panels A, B), and in Zn repletion (REC (Zn repleted cells) vs. TPEN, panels C, D).

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
