# Peer review of "Proteomic Analysis of Zn Depletion/Repletion in the Hormone-Secreting Thyroid Follicular Cell Line FRTL-5"

_nutrients, 2018, doi:10.3390/nu10121981_

Reviewer 1 Report

Review of „Proteomic analysis of Zn depletion/repletion in the hormone-secreting thyroid follicular cell line FRTL-5” by Guantario et al.

By employment of FRTL-5 thyroid cells, which are able to synthetize and secrete thyreoglobulin (Tg), express thyroid stimulating hormone (TSH) receptor, and show to extent manner TSH dependency, the authors aimed to mimic the physiological conditions of Zinc (Zn) deficiency, followed by proteomic analyses.

Proteomic analysis included Zn-depletion prior to testing and quantitative labelling of whole protein extracts with dimethyl followed my mass spectrometric analyses.

The authors demonstrated that Zn-deficiency was associated with decreased levels of Tg, which could be reversed by re-supplementation with Zn ions. Further analyses performed on Zn-depleted and control cells, revealed the modulation of 108 proteins related mainly to calcium and carbohydrate metabolism and regulation of Zn homeostasis, cell polarity, adhesion and protein translation.

Overall this is very interesting study. I would offer following suggestions/corrections?

1.       Could the authors provide any information, whether Zn-deficiency or overload may have any physiological consequences? I mean is related with any thyroid pathophysiology, disorders, cancer etc.?

2.       The first two western blot lanes presented in Fig. 1 demonstrate probably a protein-transfer problem. Could the authors provide WB image with a better quality?

3.       Since the mass spectrometric analyses were performed, it would be interesting to provide a representative MS and MS/MS spectrum demonstrating differences between controls and corresponding treatments/depletions.

4.       The authors demonstrated regulation of at least 108 different proteins? To my opinion, the most important of them should be verified my meaning of Western blot or at least on transcript level (i. e. qPCR).

Author Response

Many thanks for carefully reviewing our manuscript, please find below the itemised answers to your comments

1.            Could the authors provide any information, whether Zn-deficiency or overload may have any physiological consequences? I mean is related with any thyroid pathophysiology, disorders, cancer etc.?

Thank you for this comment. The relationships between thyroid function and Zn status is indeed a crucial aspect which has been widely reported. Relevant information is now cited in introduction (lines 65 – 68), and three more references were added in the revised version to support the statement.

2.            The first two western blot lanes presented in Fig. 1 demonstrate probably a protein-transfer problem. Could the authors provide WB image with a better quality?

The western blot experiments of the cell medium showing secreted thyroglobulin, consistently showed a very faint unclear band at the 2hr time point, leading us to believe that rather than representing a transfer problem, 2 hr growth in culture was insufficient to accumulate enough protein to be clearly visible with this technique. However, the 2 hr time point was never employed in subsequent experiments and the western blot in figure 1 is shown only to demonstrate the requirement of serum and hormones for optimal secretion of thyroglobulin. 

3.            Since the mass spectrometric analyses were performed, it would be interesting to provide a representative MS and MS/MS spectrum demonstrating differences between controls

A representative spectrum of the modulated protein HIST1H2AJ is now included as panel D in Figure 6 and three supplementary figures were also added with the corresponding raw data. In particular, the spectrum represents doubly charged ions of the peptide 22-30 of HIST1H2AJ labeled as light (TPEN) at m/z of 486.78, as intermediate (REC) at m/z of 488.80 and as heavy (CTRL) at m/z of 490.81.

4.       The authors demonstrated regulation of at least 108 different proteins. To my opinion, the most important of them should be verified my meaning of Western blot or at least on transcript level (i. e. qPCR).

While we understand the concern of the reviewer, the experiments suggested cannot be done in the timeframe for revision required by this special issue. However, although experimental validation of the proteomic data is not included, this work does not describe new pathways but rather points at a subset of proteins that were included in the Zn Proteome interaction Network (ZNP) we previously described with an in silico approach (Leoni et al 2014, full references below).  Therefore, the present study is meant as a further case study in a different tissue than that presented in this previous article. The rationale behind it is that different pathways enriched in this “overarching” ZNP are especially vulnerable to Zn-dependent modulation in different tissues and cell types, and the identification of these pathways in thyrocites can drive further research on tissue-specific processes. As stated by reviewer 2, our "set of data is a great platform to identify modulators of thyroid cell activity that are regulated by Zn" and we believe that they represent valuable information which should be available to the scientific community, to stimulate further work on the specific pathways that we indicate as Zn-modulated in thyrocites. A statement highlighting these aspects has been added to the discussion in the revised text (lines 478 - 482)

Reviewer 2 Report

The claimed goal of the manuscript submitted for review entitled “Proteomic analysis of Zn depletion/repletion in the hormone-secreting thyroid follicular cell line FRTL-5” I to provide data to support “the use of the FRTL-5 cell line as a model to investigate thyroid function”. This contribution is well written and easy to follow. However, the lack of validation and mechanistic data to understand the regulation of the proteins identified using proteomics lessons the impact of this manuscript. Essentially, this manuscript reports proteomic data showing differences in protein levels associated with Zn chelation in cells and after recovery from depletion via the addition of ZnSO4. While this work provides an interesting set of proteins modulated by Zn in thyroid cells, the authors do not verify the proteomic results using specific quantification of the proteins identified (immunocytochemistry, immunoblot) or establish a mechanistic understanding for the observed changes in protein levels. For example, a first step would be to establish if the observed increase in protein levels for a specific candidate identified in the proteomic approach is due to increased gene transcription followed by direct measurement of protein levels using standard techniques (immunocytochemistry, immunoblot, etc). Next, this line of investigation followed by experiments to identify potential transcription factors activated in response to Zn depletion/repletion. This type of experimentation would provide a deeper understanding of the role that Zn-regulated protein expression beyond the observational data provided thus providing insight into Zn-modulated thyroid function.

Co-localization experiments: The authors should include a control experiment in Figure 2 to show that the addition of TPEN eliminates the punctate fluorescence associated with Zn localization. This control is required to conclude that the observed localization is, in fact, due to the association of Zn with the reporter compound and not artifactual. The treatments similar to those used in Figure 4 should be used to obtain images. Also in Figure 2, the authors do not make it clear why only ZnT8 was chosen for Zn co-localization experiments with not the other known transporters shown in Figure 1 were not investigated; the omission of co-localization studies involving ZnT2 and ZnT4 needs to be clarified. Were experiments similar to those reported in Figure 3 also completed for ZnT2 and ZnT4?

This reviewer has no problems with the proteomic data as provided and this set of data is a great platform to identify modulators of thyroid cell activity that are regulated by Zn.

Author Response

Reviewer 2

Many thanks for carefully reviewing our manuscript, please find below the itemised answers to your comments

     1.       The claimed goal of the manuscript submitted for review entitled “Proteomic analysis of Zn depletion/repletion in the hormone-secreting thyroid follicular cell line FRTL-5” I to provide data to support “the use of the FRTL-5 cell line as a model to investigate thyroid function”. This contribution is well written and easy to follow. However, the lack of validation and mechanistic data to understand the regulation of the proteins identified using proteomics lessons the impact of this manuscript. Essentially, this manuscript reports proteomic data showing differences in protein levels associated with Zn chelation in cells and after recovery from depletion via the addition of ZnSO4. While this work provides an interesting set of proteins modulated by Zn in thyroid cells, the authors do not verify the proteomic results using specific quantification of the proteins identified (immunocytochemistry, immunoblot) or establish a mechanistic understanding for the observed changes in protein levels. For example, a first step would be to establish if the observed increase in protein levels for a specific candidate identified in the proteomic approach is due to increased gene transcription followed by direct measurement of protein levels using standard techniques (immunocytochemistry, immunoblot, etc). Next, this line of investigation followed by experiments to identify potential transcription factors activated in response to Zn depletion/repletion. This type of experimentation would provide a deeper understanding of the role that Zn-regulated protein expression beyond the observational data provided thus providing insight into Zn-modulated thyroid function.

Many thanks for this comment. We are aware that this work is only a first step toward mechanistic elucidation of transcriptional alterations occurring in Zn deficiency in thyrocites and fully appreciate the importance of further experimental work as suggested by the reviewer. However, the suggested experiments would require much longer times than the timeframe requested by the editor for revision of the manuscript and we also feel they would represent an entirely new piece of work. We believe that the strength of this manuscript is to present novel information which is not yet available in such depth on Zn metabolism in thyroid cells, which we and others can further exploit to design more specific experiments directed at clarifying the causal effect of Zn homeostasis on the pathways highlighted here. As an example, we point at the Zn containing transcription factor ZNF658, which regulates transcription of rRNAs and of several ribosomal proteins, as a potential candidate for Zn dependent regulation of proteins involved in translation in thyrocites, opening avenues for novel work with an entirely different experimental approach. This work does not describe new Zn-regulated pathways but rather points at a subset of proteins that were already included in the “overarching” Zn Proteome interaction Network (ZNP) that we previously described with an in silico approach ((Leoni, Rosato et al. 2014), full reference below), but are specifically enriched in thyrocites upon alteration of intracellular Zn homeostasis. Therefore, the present work is meant as a further case study in a different tissue than that presented in the previous article and lays the ground for further work that we and others can follow up in more detail. We thus believe that the results presented in this manuscript represent a valuable contribution that should be available to the scientific community, to stimulate further work on the specific pathways that we indicate as Zn-modulated in thyrocites. A statement highlighting these aspects has been added to the discussion in the revised text (lines 459 - 467)

     2.       Co-localization experiments: The authors should include a control experiment in Figure 2 to show that the addition of TPEN eliminates the punctate fluorescence associated with Zn localization. This control is required to conclude that the observed localization is, in fact, due to the association of Zn with the reporter compound and not artifactual. The treatments similar to those used in Figure 4 should be used to obtain images. Also in Figure 2, the authors do not make it clear why only ZnT8 was chosen for Zn co-localization experiments with not the other known transporters shown in Figure 1 were not investigated; the omission of co-localization studies involving ZnT2 and ZnT4 needs to be clarified. Were experiments similar to those reported in Figure 3 also completed for ZnT2 and ZnT4?

Co-localization experiment: we agree with the reviewer that is important to rule out artifacts in fluorescence experiments, and for this reason, we include a TPEN control in all of our Fluozin-3 assays. A representative image of FRTL-5 cells incubated with 25 µM TPEN for 2 hr with or without 25 µM ZnSO4 was indeed presented in figure 5B.

Zn transporter imaging: we chose to detail only ZnT8 intracellular localization as this transporter was previously shown to co-localize with Zn in secretory vesicles in pancreatic islet cells, and it represented therefore the most relevant Zn transporter in hormone-secreting cells. Moreover, both ZnT2 and ZnT4 were shown by others and us, to reside in vesicles distinct from Zn containing vesicles in other cell systems [(Ranaldi, Perozzi et al. 2002, Kelleher and Lonnerdal 2003), full references below]. Since the results of proteomic analysis did not show Zn-dependent differential expression of functions involved in the secretory pathway, we did not pursue further characterization of Zn transporter expression in thyrocites. To clarify this point, we have now included a comment in the discussion (lines 387-389).

     3.       This reviewer has no problems with the proteomic data as provided and this set of data is a great platform to identify modulators of thyroid cell activity that are regulated by Zn.

We thank the reviewer for appreciating the importance of the set of proteomic data reported in this manuscript as the ground for future work on Zn modulation of thyroid metabolism.

Full References:

Leoni, G., A. Rosato, G. Perozzi and C. Murgia (2014). "Zinc proteome interaction network as a model to identify nutrient-affected pathways in human pathologies." Genes Nutr 9(6): 436.

Ranaldi, G., G. Perozzi, A. Truong-Tran, P. Zalewski and C. Murgia (2002). "Intracellular distribution of labile Zn(II) and zinc transporter expression in kidney and MDCK cells." Am J Physiol Renal Physiol 283(6): F1365-1375.

Kelleher, S. L. and B. Lonnerdal (2003). "Zn transporter levels and localization change throughout lactation in rat mammary gland and are regulated by Zn in mammary cells." J Nutr 133(11): 3378-3385

Round  2

Reviewer 1 Report

I have no further comments, the authors fulfilled my objections.

Reviewer 2 Report

The previous concerns expressed by this reviewer were addressed by the authors in the revised manuscript. However, the lack of validation lessens both the interest to the readers and overall merit of the current study.